# Switching speed limits in electrically driven VO$_2$ structural Mott–Peierls transition

Alexandre Pofelski [1] ✉, Chuhang Liu[1], Spencer A. Reisbick [1], Myung-Geun Han [1], Lijun Wu [1], Henry Navarro [2,3], Erbin Qiu [2], Tianxing D. Wang[2], Shayan S. Mousavi M.[4], David J. Alspaugh[2], Marcelo Rozenberg [2,5], Shriram Ramanathan[6], Ivan K. Schuller [2] & Yimei Zhu [1] ✉

Mott materials are archetypal quantum systems actively explored as next-generation electronic and photonic platforms, with potential applications spanning non-Von Neumann computing, robotics, energy storage, and microwave technologies. Among these, vanadium dioxide (VO$_2$) has emerged as one of the most intensively studied compounds, owing to its sharp, near-room-temperature insulator-to-metal phase transition. VO$_2$ also serves as a benchmark system for testing cutting-edge theories and experimental techniques. Here, we directly visualize the electrically driven transition dynamics in VO$_2$ using a microwave-driven, frequency-tunable pulsed transmission electron microscope that combines nanometer spatial and picosecond temporal resolution. Under high-frequency (MHz–GHz) excitation, we capture the ultrafast nucleation, propagation, and dissolution of metallic domains within an operating device over millions of reversible cycles. We observe the ultrafast formation of consistent metallic nuclei beneath the electrodes, followed by the propagation of a structural phase front at 4.54 nm/ns. Our experiments show that phonon-mediated structural recovery ultimately limits reversible switching of VO$_2$ at GHz frequencies, and that a tunable regime for reversible operation spans from kHz to GHz through device engineering. Beyond VO$_2$, our approach provides a powerful framework for probing non-equilibrium structural transformations in correlated and functional materials under realistic electrical stimuli.

Mott insulators represent a class of quantum materials that possess an insulating ground state that can be switched to a conducting phase. The unique electronic properties of Mott insulators arise from the competition between electron hopping and on-site Coulomb repulsion, leading to insulating states at certain electron densities. In vanadium dioxide (VO$_2$), a well-known Mott material, the insulator-metal transition (IMT) can be triggered by external stimuli such as temperature, electric fields, or photoexcitation, resulting in structural and electronic phase changes[1–9]. This transition occurs as the insulating monoclinic phase shifts to a metallic rutile phase,

[1]Condensed Matter Physics and Materials Science Department, Brookhaven National Laboratory, Upton, New York, USA. [2]Department of Physics, University of California San Diego, La Jolla, California, USA. [3]Department of Physics, Andrews University, Berrien Springs, Michigan, USA. [4]Clean Energy Innovation Research Centre (CEI), National Research Council Canada, Mississauga, ON, Canada. [5]CNRS Laboratoire de Physique des Solides, Université Paris-Saclay, Orsay, France. [6]Department of Electrical and Computer Engineering, Rutgers, The State University of New Jersey, Piscataway, New Jersey, USA. ✉e-mail: pofelska@mcmaster.ca; zhu@bnl.gov

accompanied by lattice distortions that strongly impact the material's conductivity. Such interactions between the lattice and electronic structure are prevalent across numerous classes of quantum materials.

Ultrafast studies of $VO_2$ and similar Mott insulators have significantly advanced our understanding of their IMT behaviors. By using techniques such as pump-probe spectroscopy, ultrafast X-rays, and electron diffraction, researchers have been able to capture the IMT processes, often initiated by photons as the pump[5,10–24]. These experiments have revealed that the transition occurs through a transient monoclinic metallic state, followed by a further lattice reconstruction stabilizing into a rutile metallic state. The existence of the monoclinic metallic state is still under debate, as it may strongly depend on the structural properties of the $VO_2$ material after synthesis[20,25]. Overall, under intense optical excitation, $VO_2$ experiences an electronic and structural phase transition within tens to hundreds of femtoseconds, demonstrating its potential for high-speed switching applications.

However, these observations primarily involve photoexcitation, which, while valuable for fundamental insights, is less relevant for device applications such as neuromorphic processing or compute-in-memory technologies where electrical excitation is the main triggering mechanism[26]. A recent study demonstrated stunning structural similarities between the photo-assisted and the electrically triggered IMT in $VO_2$[16], suggesting, therefore, a similar transition mechanism through a long-lived transient monoclinic metallic phase. However, the timescale difference between the photoexcited and the electrically stimulated transitions (picoseconds vs microseconds) questions the pathways involved in the transition mechanism[27]. In addition, purely electrically driven IMTs were reported to occur around nanoseconds[28], indicating that our understanding of the IMT mechanisms remains incomplete. Gaining control over the ultrafast IMT dynamics in Mott insulators like $VO_2$ under electrical excitation, therefore, remains essential for advancing technological implementation. Numerous studies demonstrate the potential of $VO_2$ as high-frequency switches, neuronal oscillators, and sensors, justifying the strong and broad interest of the research community[29,30]. More recently, device-level thermal coupling strategies have been shown to modulate electronic functionalities, further enhancing the potential of $VO_2$-based devices[31–33]. Nonetheless, there is a knowledge gap regarding the structural information and dynamics of quantum materials such as vanadium dioxide operating under high-frequency electrical excitation.

Bridging this gap is essential not only for the design of emerging quantum technologies, but also for advancing first principles theories to model out-of-equilibrium, electrically driven correlated electron systems[34]. Resolving the propagation of the IMT phase front requires simultaneous high spatial and temporal resolution under operando electrical excitation. While modern ultrafast electron microscopy (UEM) provides the necessary spatiotemporal resolution, its electric excitation frequencies are limited to several orders of magnitude below those relevant for real-world device operation, particularly in wireless and quantum communication applications. To overcome this limitation, we have developed a purely electrical pump-probe ultrafast electron pulser, eliminating the need for laser-based excitation, that enables access to the 10 GHz frequency regime with high transmission efficiency (Fig. 1). Its field emission electron source is synchronized with the sample placed in a custom-designed RF sample holder through a frequency-tunable RF cavity, allowing for precise, time-resolved high-frequency excitation and observation. Utilizing a single-electron-sensitivity direct electron detector, we can capture the extremely weak signals associated with the propagating phase front. This advanced setup enables direct visualization of the electrically induced phase transition dynamics with picosecond temporal and nanometer spatial resolution.

In this study, we investigate the ultrafast structural dynamics of a $VO_2$ thin film fabricated as a device under typical operational conditions up to several GHz frequencies. Using ultrafast electron diffraction, we probe the structural phase transition of $VO_2$ across MHz to GHz frequencies and reveal a frequency-dependent response of the phase transformation. This dependence defines fundamental lattice constraints for reversible switching operations that are essential for all envisioned applications. We further perform direct imaging of the domain formation and dissolution at 1 MHz using ultrafast transmission electron microscopy, providing a nanoscopic visualization of the Mott switching front inside the device channel. Our findings delineate the operational limits for electrically driven prototypical Mott switching devices and establish a methodology generalizable to emerging materials wherein electronic functionalities are closely tied to lattice dynamics.

## Results and discussion
### Frequency dependent structural phase transition

To probe the frequency-dependent behavior of the $VO_2$ device, we used the purely electrically driven ultrafast electron pulser[35] (Fig. 1) offering frequency tunability over several orders of magnitude. The pulser chops the continuous electron beam from the electron gun into a train of electron pulses at a selected frequency. The all-electrical configuration simplifies the synchronization between the electron probe and the sample excitation, allowing us to study electrically induced ultrafast dynamics from 1 Hz to 10 GHz. The ultrafast dynamics are recorded stroboscopically (requiring complete reversibility) through signal-matched triggers passing between each signal generator, allowing the pulsed electron beam (probe) to observe the sample response to the electrical excitation (pump) at different phase delays (i.e., time-points). In this study, we investigate the reversible structural phase transitions in a $VO_2$ device using a modified radio frequency (RF) excitation technique, previously employed to elucidate spin-wave propagation under RF field[36]. Additionally, we developed a methodology utilizing MHz AC to specifically explore purely electrically induced phase transitions in the radio wave regime[37].

The geometry of the $VO_2$ device grown on a sapphire substrate is presented in Supplementary Fig. S1a, b. The $VO_2$ sample is a textured polycrystalline film of 600 nm thick with a typical metal-insulator transition at 67 °C (more information in Supplementary Fig. S1c, d) and is connected laterally with two electrodes. A high-frequency electric potential is applied to an electrode, and the structural evolution of the $VO_2$ material is characterized. The two-terminal device is, here, observed in cross-section, allowing the device operation to be also characterized along the film depth. As $VO_2$ experiences a structural phase transition from a monoclinic to a rutile crystal symmetry, the structural evolution of $VO_2$ can be probed using Selected Area Electron Diffraction (SAED) and by tracking the presence of the monoclinic signature associated with the V−V dimerization (details presented in Supplementary Fig. S2). Owing to the frequency flexibility of the instrument, the ultrafast structural changes are measured using a 1 MHz pulse train profile and a 4 GHz radio frequency (RF) signal as presented in Fig. 1. Both are relevant options for high-frequency applications and are extensively investigated in phase change materials. The key challenge is to directly characterize the frequency dependance of the $VO_2$ structural phase transition as shown in the schematic diagram in Fig. 2e and define experimentally the boundaries between different modes of operation.

Electron diffraction patterns are highlighted in Fig. 2a, b at different states of the electrical excitation from the selected area circled in Fig. 3a (approximately 400 nm in diameter and positioned below the left electrode of the $VO_2$ device). The results confirm that both the MHz pulse train and RF microwaves trigger the structural phase transition as the monoclinic diffraction peaks fade out when the excitation is turned on (complete datasets available in supplementary movies 1,

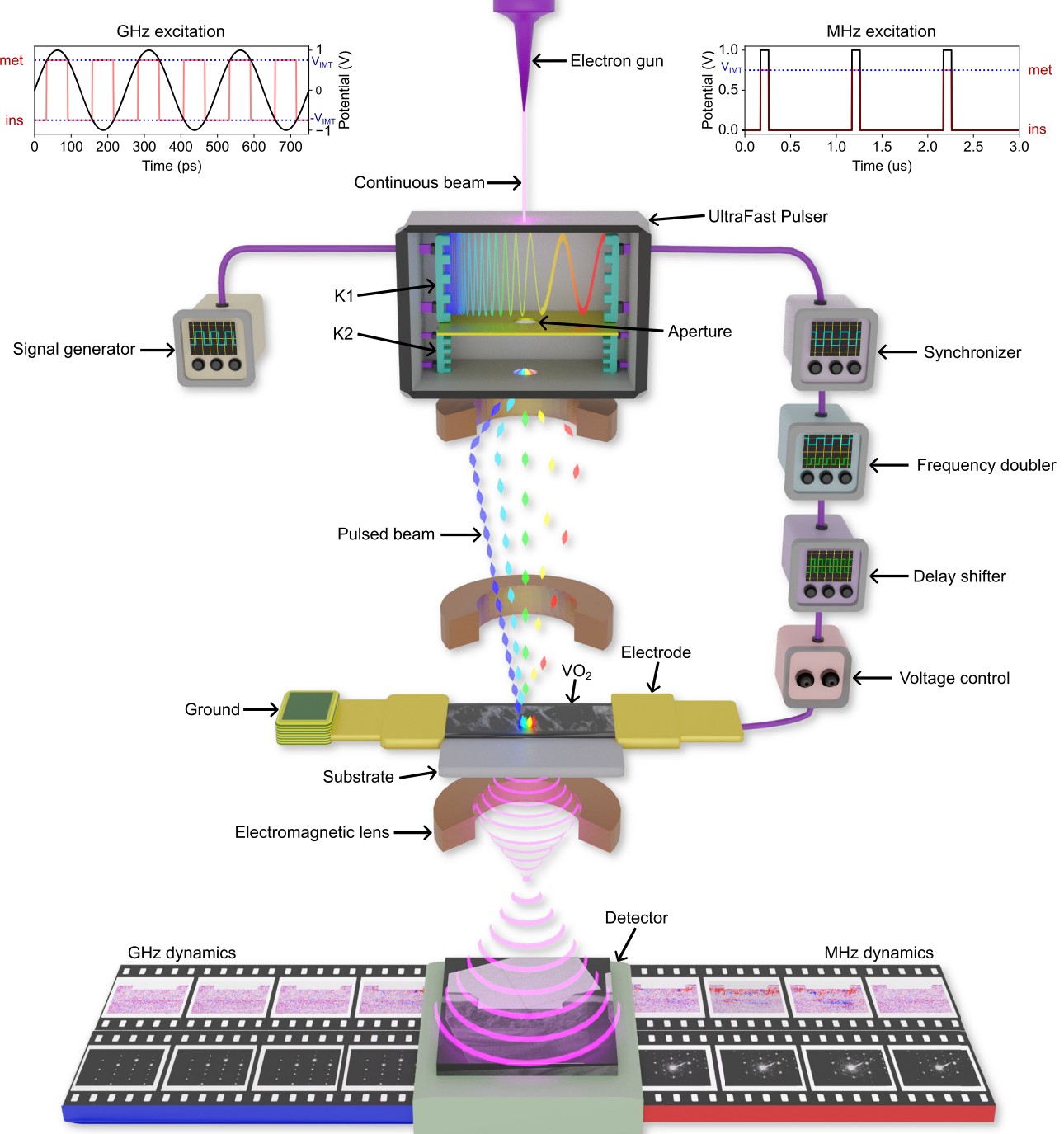

**Fig. 1 | Illustration of the pure-electrically driven ultrafast transmission electron microscopy experimental setup to study a VO₂ two-terminal device.** The ultrafast pulser modifies a conventional continuous electron beam into a frequency-tunable pulsed electron beam. The pulsed electron beam (probe) is synchronized with the electrical stimulation (pump) sent into the VO₂ film grown on a sapphire substrate and connected laterally with two electrodes to the electric circuit. Two frequency operation modes are depicted with their respective pump profiles: a 1 MHz square wave train and a 4 GHz sinusoidal excitation, both crossing the IMT threshold (more details in the methods section). The red profiles correspond to the ideal case of instantaneous Mott switching between the metallic (met) and insulating (ins) states following the voltage profiles.

2). Measuring the intensity of the monoclinic signature over time, a frequency-dependent behavior is clearly observed. For the MHz experiment, the intensity of the diffraction peaks follows, with some delay, the excitation profile, indicating a dynamic behavior oscillating between the monoclinic and rutile states. From the data, the time constant for the monoclinic to rutile structural transition is estimated to 36 ns +/− 10 ns and to 107 ns +/− 21 ns from the rutile to monoclinic phase recovery. For the GHz case, the monoclinic signal is absent and

remains constant regardless of the phase of the RF. Under GHz excitation, no structural dynamics are observed, and the VO₂ remains locked in its rutile metallic state. To determine whether the absence of structural dynamics extends beyond the SAED measurement, we conducted an in-situ RF 4D-STEM experiment at 4 GHz (Supplementary Fig. S3). By mapping the distribution of monoclinic-rutile phases across the entire device, we confirm that the VO₂ is fully switched into the metallic phase. It is worth pointing out that several regions near the

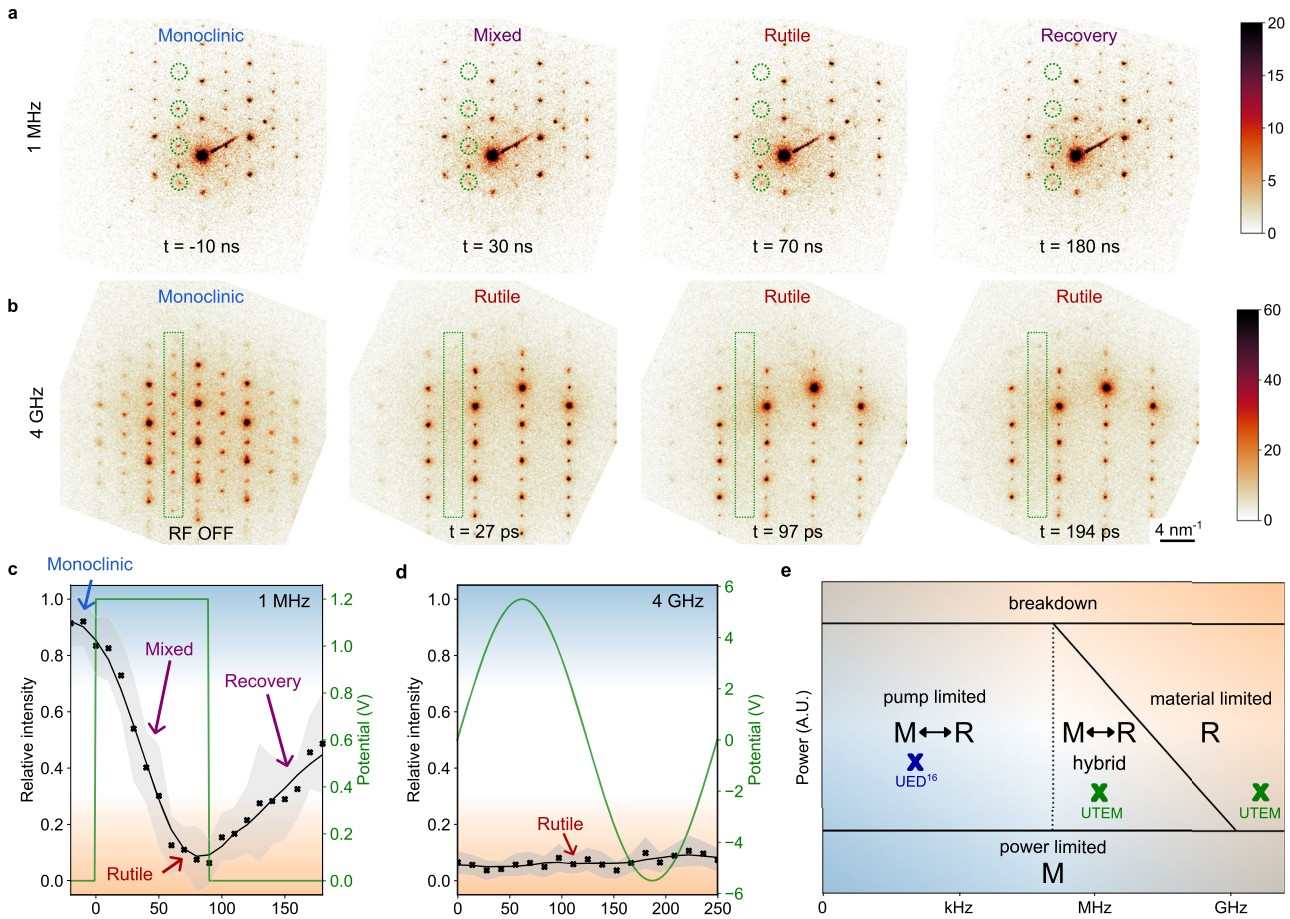

**Fig. 2 | Ultrafast Selected Area Electron Diffraction on the VO₂ thin film with both MHz and GHz excitation. a**, **b** Electron diffraction patterns at different times under the 1 MHz pulse train excitation and under the 4 GHz radio frequency excitation, respectively. The color scale represents electron counts and is saturated for better visibility of the weak diffraction peaks. Several diffraction peaks related to the monoclinic distortion are marked in green dashed circles and boxes (more information in Supplementary Figs. S2, S4). **c**, **d** Averaged relative intensity of the monoclinic diffraction peaks with time overlapped with the excitation profile at

1 MHz and 4 GHz, respectively. The gray band represents the standard deviation of the relative intensity at each time point. **e** Schematic of the structural switching dynamics diagram with power and frequency highlighting different modes of operation. M and R refer to the monoclinic and the rutile crystal structure, respectively. The green marks correspond to the two frequencies investigated in this study, and the blue mark to the electrically stimulated Ultrafast Electron Diffraction (UED) study reported on a similar two-terminal VO₂ device[16].

substrate stay monoclinic under GHz excitation and do not experience structural switching with RF.

While the structural transition from monoclinic to rutile has been reported to be in the picoseconds timescale, the recovery of the monoclinic from the rutile state is expected to be at a different order of magnitude[13]. Indeed, once the photo or the electrical excitation is removed, heat needs to be dissipated from the rutile phase to recover the monoclinic structure, and VO₂ has a relatively low thermal conductivity of 6 W/mK. Therefore, it is expected that a frequency threshold exists inhibiting the concomitant structural switching with the electric excitation profile, delineating a material-limited region in the switching phase diagram in Fig. 2e. Our experiments establish that for VO₂, the limit for structural switching using electrical stimulation is in the GHz frequency setting, the ceiling for high-frequency operations. It is interesting to note that the complete transition from the monoclinic to the rutile crystal structure is slower electrically than with photo excitation (around tens of nanoseconds in our study). Such behavior has been noted in the literature with an even longer time-frame in microseconds[16]. As the electron diffraction patterns in Fig. 2a, b are averaged over the selected area, the slow fading of the monoclinic diffraction peaks can be interpreted as a slow removal of the V-V

dimerization or as a mixture of monoclinic and rutile phases slowly evolving in the selected area. Without considering the structural phase front propagation, results from diffraction alone are insufficient to address the fundamentals of the electrically stimulated metal-insulator transition. Direct imaging is therefore important and is discussed in the following.

## Imaging metallic domains at MHz frequency

Identifying thermal dissipation as a fundamental constraint for high-frequency switching is one essential piece of information, but it doesn't provide a complete understanding of the VO₂ device functionalities at high frequencies. The VO₂ thin film is indeed integrated on a substrate and connected to two electrodes (acting as heat sinks) and has a dedicated geometry directly influencing the heat dissipation process. To gain more insight into the functional operation, the ultrafast experiment on the same device is repeated in real space using 1 MHz pulse trains.

As the imaging contrast in bright-field TEM is not solely due to the change of crystal structure, the direct interpretation of the images is not straightforward. Nonetheless, if the sample thickness and the grain arrangement remain nearly the same under electrical excitation, a

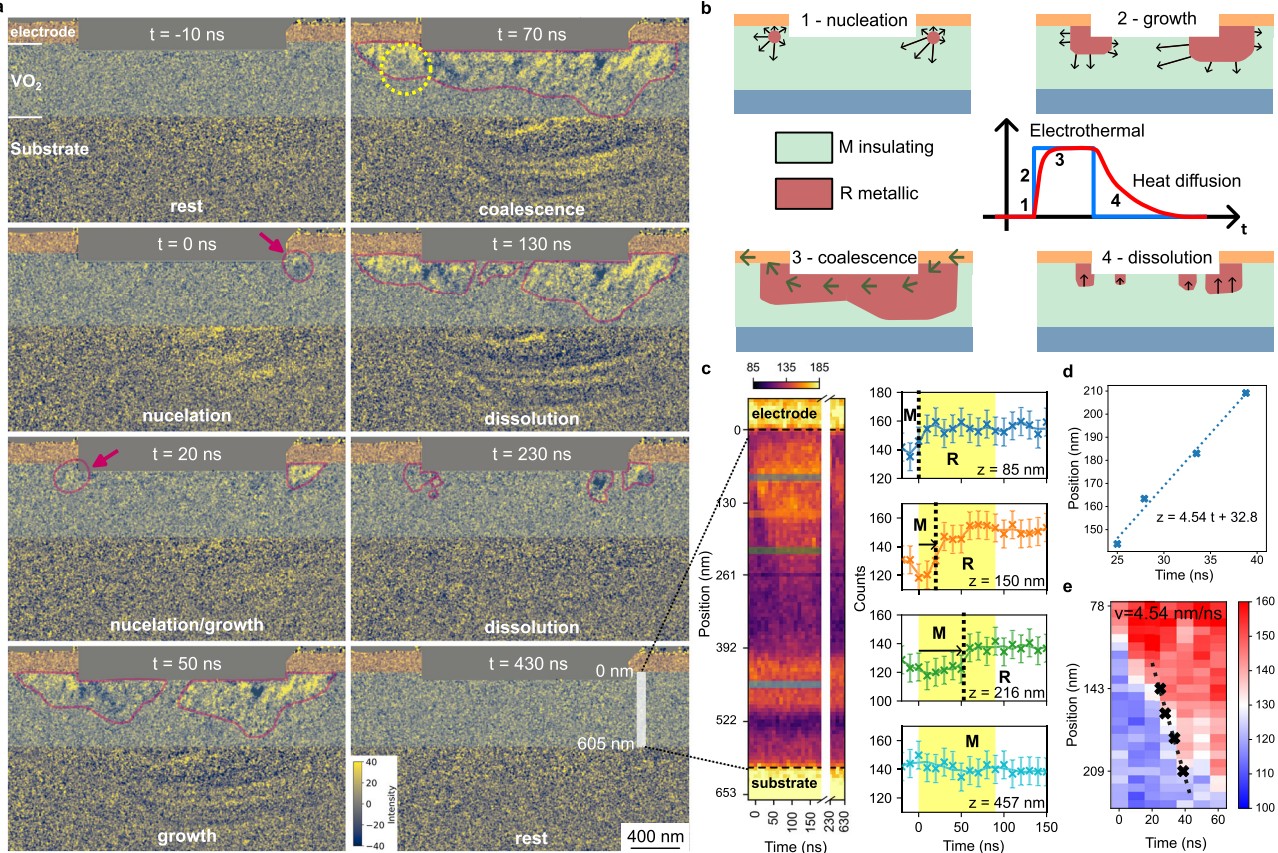

**Fig. 3 | Ultrafast Bright-Field Transmission Electron Microscopy (BF-TEM) of the VO₂ two-terminal device. a** Difference images between two BF-TEM images using the time stamp t = −70 ns as the reference. The areas in orange and green highlight the electrode and the VO₂ region, respectively. The yellow dotted circle corresponds to the SAED region from Fig. 2. The purple contours delimitate the metallic domains in the VO₂ film. **b** Illustration of the four phases of the metallic domain dynamics. **c** Space-time intensity plot along the vertical line profile marked at t = 430 ns representing the local variations of the BF-TEM intensity with time. Several horizontal line profiles at different depth positions are presented on the right, identifying, with the vertical black dotted line, the time when the monoclinic

M to rutile R structural transition occurs. The yellow section in the line profiles highlights the 90 ns time frame when the electrical pump is at 1.2 V. The error bars correspond to the standard deviation of the beam intensity in vacuum. **d** Linear fit of the IMT time occurrences during the growth phase function of the depth position (more information in Supplementary Fig. S5). **e** Representation of the linear fit from (**d**) on a small section of the space-time intensity plot from (**c**), relevant for the growth phase. The slope of the linear fit corresponds to the structural phase front velocity during the electrical excitation. The color scales in (**c**, **e**) represent electron counts and are saturated for a better visualization of small changes in contrast.

change of contrast at different time frames can be qualitatively related to a structural transformation. Variations in the diffracted beam intensities caused by the structural transition to another crystal symmetry are indeed expected to notably alter the contrast in Bright Field Transmission Electron Microscopy (BF-TEM) imaging[38]. Since the VO₂ film is polycrystalline, each grain undergoes a distinct change of intensity in BF-TEM imaging during the structural phase transition, meaning that the absolute intensity itself cannot be used to track the structural transformation. However, the difference of bright-field TEM images using the materials' state at 0 V as a reference (Fig. 3a) is robust to the intensity variations from the VO₂ texture. Thus, any region in the VO₂ film exhibiting a change of intensity above the noise level is interpreted as having undergone a structural transition to the rutile phase, irrespective of the sign and the magnitude of the intensity difference. Capturing the difference images at different time points reveals the dynamical formation and dissolution of the metallic domains, mapping the device's operation in time.

Videos of the ultrafast structural dynamics are available in supplementary movies 3, 4, and a summary of the experimental results is presented in Fig. 3a. Due to the significant noise level in the data, a single threshold cannot reliably distinguish the rutile regions from the monoclinic phase. To aid visualization, the intensity in the difference

images shown in Fig. 3a and in supplementary movies 3, 4 has been mildly saturated to highlight the metallic rutile regions. Further denoising and image filtering procedures are provided in the Supplementary Figs. S6–8, to facilitate the identification of rutile metallic domains in the VO₂ film at each time frame. The selected snapshots in Fig. 3a represent the different states of the metallic domains dynamics through a complete cycle (illustrated schematically in Fig. 3b): the rest state with no metallic domains (VO₂ entirely monoclinic), the nucleation phase of nanoscale rutile domains just beneath the electrodes (circled in purple in Fig. 3a), the growth from the nucleation sites with a slight lateral preferential direction and a penetration towards the substrate direction, the coalescence of the filament between both electrodes and stabilization, and the dissolution of the filament with the recovery of the monoclinic structure. As the ultrafast experiment is performed under stroboscopic conditions, only the coherent, repetitive and reversible processes are experimentally captured. Each snapshot in Fig. 3a represents the cumulative results after 5 million electrical cycles, demonstrating the repeatability of VO₂ device cycling despite the intrinsic stochasticity of Mott insulators. The striking reproducibility arises from the precise control of metallic channel formation and dissolution provided by the high-frequency electrical excitation, distinguishing it from ultrafast photoexcitation processes.

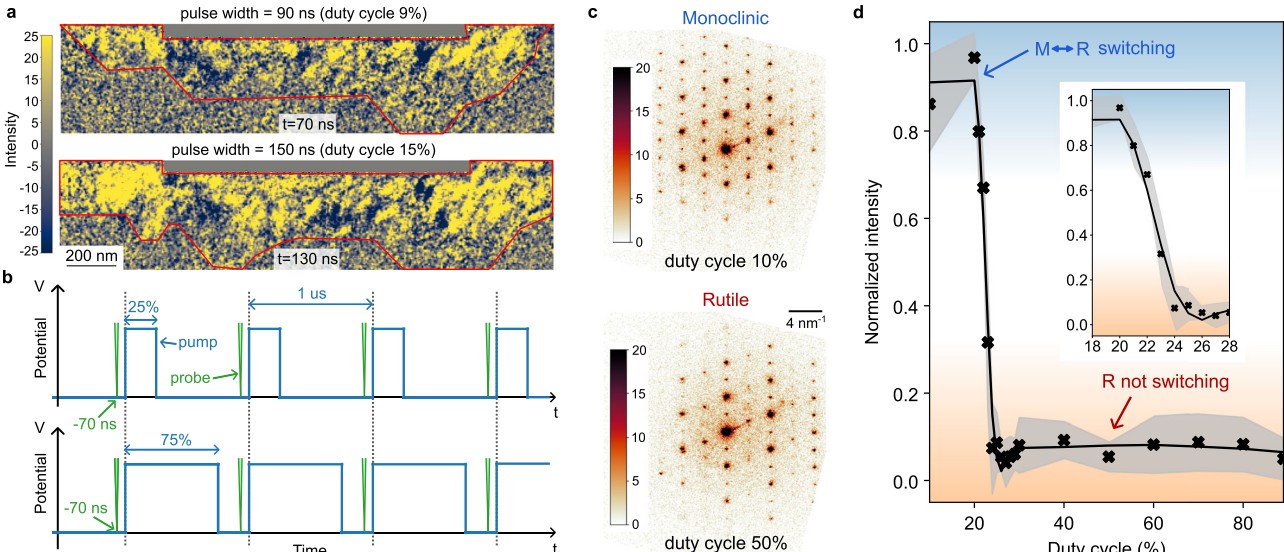

**Fig. 4 | Impact of the electric pump duty cycle on the VO₂ monoclinic phase recovery. a** Difference images as in Fig. 3a cropped on the VO₂ area only, showing the metallic filament once stabilized at two different pulse widths of the 1 MHz pulse train. The color scale is strongly saturated to better observe small abrupt changes of contrast. The boundary of the filament is qualitatively highlighted in red. **b** Schematic of the experimental 1 MHz square train excitation with varying duty cycle, highlighting the stroboscopic imaging configuration at t = −70 ns. The pulsed probe is fixed, and only the pulse pump width is varying, **c** Electron diffraction patterns from the same location with different duty cycles. The color scale represents electron counts and is saturated for better visibility of the weak diffraction peaks. **d** Relative intensity variation of the monoclinic diffraction peaks with pulse width. The inset presents a magnified view of the abrupt transition. The gray band represents the standard deviation of the relative intensity at each duty cycle.

Comparing the results from Fig. 3a with the electron diffraction patterns in Fig. 2a, it is possible to complement the interpretation of the fading monoclinic diffraction peaks. While the nucleation sites appear instantaneously (faster than the probe resolution of 2.2 ns), the propagation of the rutile phase front qualitatively follows the time scale in Fig. 2a. Our observation suggests that the fading is due to the structural wave front propagation from the nuclei, causing a structured coexistence of monoclinic and rutile phases within the selected area. In addition, examining the time stamps in Fig. 3a reveals that the growth phase occurs faster than the dissolution and agrees with the ultrafast SAED results from Fig. 2a, pointing again to phase coexistence.

Focusing on the growth phase, the velocity of the structural wave front can be measured using the space-time intensity plot in Fig. 3c and the line profiles at different depths of the VO₂ film. An abrupt change of intensity along the time axis corresponds to the structural change from the monoclinic to the rutile crystal structure. Notably, the time required for the transition to occur increases with the depth of the VO₂ thin film, as shown by the arrows in the line profiles in Fig. 3c for z = 85 nm, z = 150 nm and z = 216 nm. Beyond approximately 400 nm, no structural transition is observed (as highlighted in Fig. 3c for z = 457 nm), and the VO₂ remains monoclinic down to the substrate interface. The time delay between electrical stimulus and the actual structural phase transition was extracted as a function of depth using processing routines detailed in Supplementary Fig. S5 and linearly fitted as shown in Fig. 3d. The linear fit traces the monoclinic to rutile transition contour of the space-time intensity plot in Fig. 3e during the growth phase. Therefore, the slope of the linear fit corresponds to the structural wave front velocity, estimated experimentally to 4.54 nm/ns.

Additional insight can be obtained from the substrate response itself, as ripple-like contrast fluctuations are visible in the sapphire region in Fig. 3a, concomitant with the structural dynamics of the VO₂ film. These contrast variations arise from the motion of bending contours, indicating elastic distortions of the substrate induced by the VO₂ structural phase transition. As VO₂ undergoes a slight volume change upon switching to the metallic phase, a mechanical constraint is imposed on the underlying sapphire, which is expected to respond elastically. No measurable delay is observed between the nucleation of the metallic phase at the top of the VO₂ film and the initiation of bending contour motion in the substrate. This suggests that, within the temporal resolution of the experiment (2.2 ns), mechanical stress propagates from the nucleation site into the substrate without inducing a structural transformation of the VO₂ region in between. While this observation confirms the stress transfer across the film-substrate interface, it also indicates that mechanical contributions are not the dominant factor governing the propagation of the IMT structural front in VO₂.

Overall, a phase front velocity in the order of nanometers per nanosecond, as measured in Fig. 3e, is significantly lower than both the drift velocity of electrons in solids and the speed of ultrasound in crystalline materials. Consequently, neither a purely electrical nor a mechanical model can fully explain the Mott structural dynamics as observed in photoexcitation[18]. Heat waves are typically in the order of nanometers per nanosecond and depend on the thermal gradient, the materials used and the device geometry[39]. Based on our experimental data, the structural wave front evolution seems to be mostly driven by non-equilibrium thermal transport with assistance of the electric field contributing to the nucleation and growth phases (electrothermal mechanism), rendering support to recent calculations[34]. Thermal management is therefore of fundamental importance for the VO₂ IMT propagation and imposes constraints on the non-linearity of the device functionalities (such as the time for firing or for recovery in a neuromorphic processing unit).

### Stimulus-dependent insulating phase recovery

The impact of the thermal budget on the VO₂ device can be observed by changing the voltage profile and increasing the pulse train width (or the duty cycle defined as the fraction of time the voltage is on over a period). As the potential remains high for longer durations, the current flow between the two electrodes is also maintained for longer,

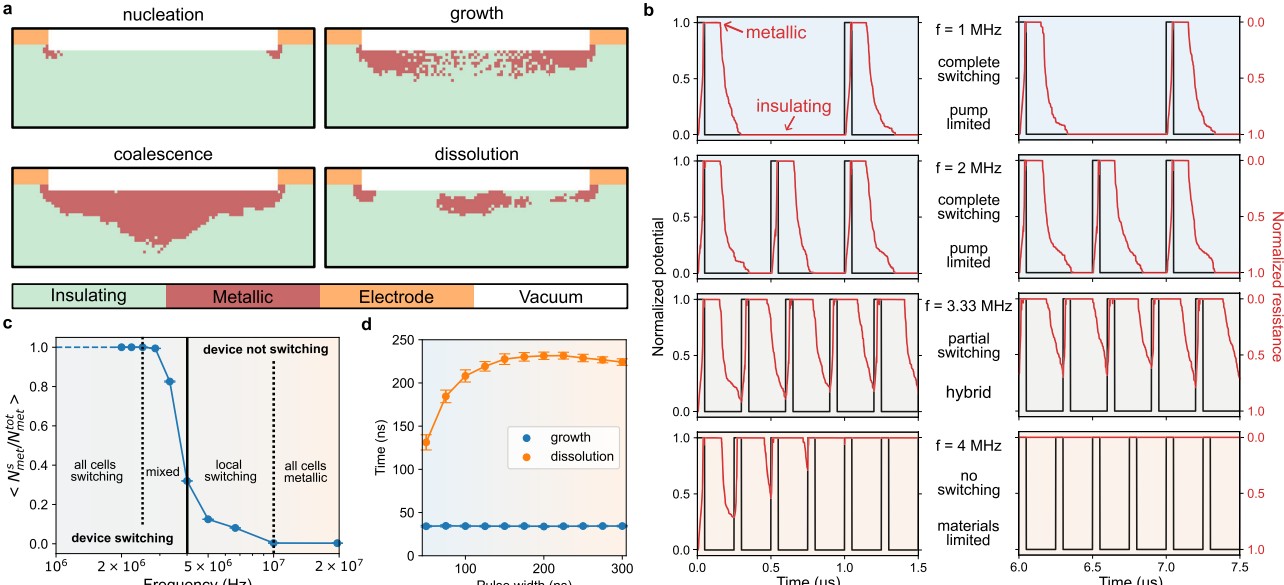

**Fig. 5 | Simulation of the electrically stimulated VO$_2$ two-terminal device using a Mott Resistive Network. a** Simulated phase mappings (insulating and metallic VO$_2$) at different stages of the structural dynamics (additional details in the supplementary information). **b** From top to bottom, normalized resistivity profiles of the VO$_2$ device with a 50 ns pulse width voltage profile at 1 MHz, 2 MHz, 3.33 MHz and 4 MHz, respectively (frequency scaled from the 1 MHz ultrafast experiment). **c** Ratio of the number of metallic cells recovering their insulating state ($N_{met}^s$) over the maximum number of metallic cells ($N_{met}^{tot}$) after one voltage cycle (50 ns pulse width at different frequencies). The average of the $N_{met}^s/N_{met}^{tot}$ ratio is reported and represents the fraction of the metallic filament that is structurally switching. **d** Relation between the metallic filament growth and dissolution time with pulse width, estimated from the simulated resistivity profiles through a single voltage pulse.

increasing the thermal contribution through the Joule effect. Videos of the ultrafast structural dynamics with a pulse width of 150 ns are presented in supplementary movies 5, 6 and can be compared to the previous experiment using a pulse width of 90 ns. Results in Fig. 4a highlight the stabilization of the metallic filament at two different pulse widths (90 ns and 150 ns). A wider expansion of the filament towards the substrate is noticeable with a larger duty cycle. As a result, the filament dissolution takes significantly longer, so as the recovery of the VO$_2$ monoclinic state. The metallic domains dissolution, driven by thermal dissipation, is clearly the limiting factor for the structural cycling behavior of the Mott device.

The limit of the structural cycling behavior can be evaluated using the ultrafast transmission electron microscope in an unconventional configuration as highlighted in Fig. 4b. The stroboscopic pulsed electron probe is positioned just before the pump rises and the structural state of the VO$_2$ is captured using the same SAED method as shown in Fig. 2. The pulsed probe is fixed in time, so the ultrafast experiment is not time-resolved. However, the experiment carries temporal information, as the device is expected to be in its resting state within this timeframe. Any changes in the diffraction pattern are an indication of the non-congruent recovery of the original monoclinic state. Examples of pulsed electron diffraction patterns are shown in Fig. 4c at two different duty cycles and highlight a different intensity distribution of the monoclinic diffraction peaks (the entire dataset is available in supplementary movie 7). The monoclinic recovery over cycling is described in Fig. 4d by tracking the monoclinic diffraction peaks intensity as in Fig. 2 at different duty cycles. For a low duty cycle, the monoclinic crystal structure is observed as the full monoclinic recovery is expected. At a larger duty cycle, the diffraction pattern highlights the rutile crystal structure, demonstrating that the material does not recover the monoclinic phase during the 1 MHz cycle and remains in the rutile phase as in the GHz experiment. At a duty cycle of 23%, the results reveal a coexistence of monoclinic and rutile phases, indicating either a spatially distributed mix of phases within the selected area and/or a temporal mixing of phases over successive cycles. It is

interesting to notice that even for relatively short duty cycles, the complete recovery of the monoclinic VO$_2$ is not guaranteed. Therefore, in addition to the electrical potential frequency and amplitude, the pump profile also directly influences the device's high-frequency functionality.

## Frequency domains for VO$_2$ Mott devices

To explore the influence of the voltage frequency and pump profile of the electrically stimulated VO$_2$, the two-terminal device is simulated using a Mott Resistive Network (MRN) model[34,40]. The phenomenological MRN model has been demonstrated to capture the electric cycling behavior and the metallic domain formation dynamics of Mott insulator devices on planar structures[34,40,41]. It is important to point out that the MRN simulation employs arbitrary units; therefore, for the sake of comparison, the reported data is scaled in both space and time using experimental data, and metrics such as resistance and voltage are normalized. As the device geometry impacts the thermal distribution, the two-terminal device prepared as a thin lamella of approximately 150 nm thick must be considered for the MRN simulation. The electron transparent device is indeed in a different configuration than in its planar setup. Despite the thickness limitation necessary for the TEM characterization, the VO$_2$ device is still embedded with two electrodes and a substrate in the vicinity acting like heat sinks (like in the planar device) and therefore keeps its main functionalities without additional thermal management[42].

By adapting the device geometry into a two-dimensional model in cross-section and using the ultrafast experimental results at 1 MHz as a reference point, the different phases of the structural dynamics were successfully reproduced as shown in Fig. 5a (and in supplementary movie 9). The electrothermal picture from the MRN model qualitatively correlates with the structural dynamics observed experimentally. Some small differences remain, such as a more pronounced vertical direction for the structural phase front. We believe the stronger penetration along the depth observed experimentally is due to the texture of the VO$_2$ film having preferential grain orientation and

therefore resulting in an anisotropic structural wave front propagation (additional information about the grain arrangement is available in the Supplementary Fig. S9). Nonetheless, the electrothermal model, commonly applied in DC Mott switching, is extensible to higher frequencies from a structural perspective and differs from the phase-field model used with photoexcitation.

Complementary to the domain formation dynamics, the resistivity can also be extracted from the MRN model at different time steps of the cycling voltage. Results are presented in Fig. 5b for different frequencies using the same pulse width. Under such simulation conditions, the filament dissolution process is the limiting factor for the resistive switching and can be totally suppressed at high frequencies. At moderate frequency, a partial switching regime is revealed that is sensitive to some stochasticity. Indeed, the MRN model considers on each cell a probability distribution for the insulator-metal transition, based on the local temperature, that impacts the global resistivity profile during cycling. This additional stochasticity mimics to some extent the variability in neuronal activity, and the dissolution effect has interesting implications, for example, in neuromorphic short-term learning and plasticity. Another view of the impact of frequency on the resistive switching is highlighted in Fig. 5c by looking at the ratio of the filament cells cycling with frequency. It is possible to notice that some local switching persists at moderate frequency, showcasing another contribution of the local stochasticity despite the device not switching globally. Exploring the effect of the pulse width (or duty cycle) in Fig. 5d, the time difference between the growth and dissolution dynamics corroborates the ultrafast structural results. In addition, the simulations suggest that the dissolution time increases with pulse width (without affecting the growth time) and then saturates, indicating an increase and stabilization of the thermal budget. The MRN model points out the advantages of using short pulse widths when thermal aspects are critical. Additional electrical measurements performed at moderate frequency on a VO$_2$/sapphire device in planar configuration are presented in Supplementary Fig. S10 and confirm the impact of the thermal budget on the recovery of the high-resistivity state (monoclinic phase). These measurements further demonstrate that high-frequency cycling is electrically limited by the metallic dissolution process, emphasizing the critical role of pulse shape and temporal profile when increasing the operating frequency as suggested by the MRN model.

Combining simulations and experimental results, it is possible to identify distinct modes of operation for devices based on the structural switching dynamics of quantum materials such as VO$_2$ and discuss the phase diagram presented in Fig. 2e. In the power range where the Mott switching is possible without electrical breakdown, the electrical and structural switching is concomitant at low frequencies, and the electrical excitation defines the operation envelope of the device. For high frequencies (above the GHz range for VO$_2$), the structural switching is suppressed by the material itself, which is not able to recover its initial crystal structure during cycling. In contrast, at moderate frequencies (around the MHz range for VO$_2$), a hybrid mode of operation is anticipated, offering new opportunities for device engineering and dynamic phase control. Depending on the switching material, the substrate, the device geometry, and the stimulus profile, it is possible to both conserve or mitigate the structural switching and utilize further the stochasticity inherently present in Mott insulators[32,43]. Navigating through the structural switching phase diagram, as shown in Fig. 2e, should aid in choosing and designing the basis of hardware for artificial intelligence or photonics. It is important to note that our study focuses on Mott insulators based on structural switching. Systems undergoing isostructural Mott transitions may not exhibit the same behavior or be subject to the limitations discussed here.

By employing our newly developed microwave-based purely electrical ultrafast electron microscopy (featuring electron-pumping and electron-probing), we directly image the electrically driven dynamics of a VO$_2$-based resistive switching device. Our results establish a definitive frequency threshold for reversible switching, revealing fundamental speed limits for emerging Mott device technologies. Below the frequency threshold, we resolve the complete ultrafast structural evolution over a full life cycle, from nucleation, growth, and coalescence to eventual dissolution. The structural phase front velocity is measured to be approximately five nanometers per nanosecond, providing a fundamental design parameter for VO$_2$-based devices. Notably, we observe that structural switching is fully suppressed when the voltage pulse width exceeds a specific limit. These findings highlight that the kinetics of phase reversal impose the ultimate operational frequency ceiling for Mott insulators undergoing coupled electronic and structural transitions.

## Methods

### VO$_2$ thin film synthesis

VO$_2$ films with a thickness around 600 nm were deposited on R-plane sapphire substrate using RF magnetron sputtering with a base chamber pressure of $1.25 \times 10^{-7}$ Torr. Prior to deposition, the substrate was ultrasonically cleaned in acetone, methanol and isopropanol, each for 10 min to remove organic contaminants. An in-house V$_2$O$_3$ sputtering target fabricated with stochiometric V$_2$O$_3$ powder (99.9% purity) was used for sample growth. The deposition was conducted in a gas mixture of 3.8 mTorr Ar and 0.4 mTorr O$_2$ while the substrate was held at 400 °C. Upon finishing the deposition, the sample was cooled to room temperature at a 12 K/min rate in the same gas mixture to allow oxidation of the deposited film to transition into VO$_2$. The structural properties of the fabricated films were characterized using a Rigaku Smartlab 3 kW X-ray diffractometer and showed a textured polycrystalline VO$_2$ (see Supplementary Fig. S1a). The electrical transport was measured using a Keithley 2450 source meter with a current range of 5 uA – 5 nA in a probe station. The VO$_2$ films showed more than 3 orders of magnitude change in resistance across the IMT (see Supplementary Fig. S1c).

### Device fabrication

To fabricate a two-terminal device suitable for Ultrafast Transmission Electron Microscopy characterization, a layer of Ti (around 100 nm) was first deposited on top of the VO$_2$ thick film grown on sapphire using the DC sputter method (Kurt J. Lesker PVD 75 sputter). Then, an in-situ lift-out process was performed using the Helios G5 Dual Beam operating at 30 keV with an 8 keV clean-up. The sample was left intentionally thick (around 150 nm) to increase its robustness during the in-situ pulsed biasing experiment. The sample was then rotated by 90° and transferred using a nanomanipulator into a Norcada SiN membrane chip (10 µm-window) with home-designed Ti/Au signal lines patterned with e-beam lithography and e-beam evaporation deposition. The VO$_2$ device was finally welded to the chip using the ion-beam-assisted carbon deposition inside the Helios G5 and connected to the signal line with ion-beam-assisted Pt deposition. A final FIB milling step was performed to remove the top Ti layer between the two electrodes and form a two-terminal device.

### Microwave-based ultrafast transmission electron microscopy

All microwave-based (electron-pumping and electron-probing) ultrafast transmission electron microscopy experiments were performed on a modified JEOL JEM-2100F TEM with a frequency-tunable ultrafast pulser between the emission chamber and the first condenser lens. The pulser was developed through a collaboration between BNL and Euclid Techlabs, with the support of US DOE SBIR grants. The pulser is composed of two twin-metallic strip lines, designated K1 and K2 (in the order the electron beam passes through), which transmit electrical signals that modulate the electron beam across a chopping aperture to create electron pulses. Time-resolved experiments were performed in

two different equipment architectures, namely methodologies to encompass the MHz and GHz regimes. Additionally, the time series acquisitions were randomized to decorrelate the observable dynamics from experimental artifacts.

**Low frequency regime.** For experiments requiring excitations of frequencies up to 1 MHz, the microscope is set up in the fast-chopping[37]. Here, a 40 V DC bias was applied to one side of K1 and 40 V pulses of 2.2 ns were applied to the opposite side at a 1 MHz frequency from a high voltage pulse generator (AV Tech). The DC bias deflects the electron beam off the optic axis, while the opposing voltage pulses momentarily (2.2 ns) position the beam through a 25 um chopping aperture to create the electron pulses. The 1 MHz signal creating the electron pulses is used to synchronize an alternate pulse generator which supplies the periodic excitation arriving at the sample (5 ns rise time and 1 MHz frequency). The amplitude of the electric signal was raised sequentially to 1.2 V until the observation of a structural change in the material under continuous beam. Experiments were repeated using pulse widths of 90 and 150 ns to observe the difference in growth of the monoclinic phase. The relative time delays were controlled from the AV tech module in terms of nanoseconds.

**High-frequency regime.** For experiments requiring excitations above 1 GHz, the microscope is set up in the sweeping mode[35,37]. Here, two 2 GHz radio frequency (RF) transmissions were applied to both sides of K1, being 180° out-of-phase, meant to amplify the sweeping motion of the electron beam. Another set of identical 180° out-of-phase 4 GHz RF signals was applied to K2 to correct the sweeping modulation of the electron beam (see Fig. 1). As a few-millimeter gap exists between K1 and K2, a slight phase shift of K2 is required compared to K1 to account for the additional travel time of the electron beam. Next, the traveling RF was output from K1 and passed through isolators, amplifiers and attenuators for power control. The signal was frequency doubled before being sent to the sample, and a band-pass filter was used to remove additional harmonics in the signal. The power and excitation quality were measured at the entrance of the sample holder using a Tektronix spectrum analyzer capable of measuring powers up to 25 dB and 6 GHz. Power fluctuations in the phase delay shifter were monitored by measuring baseline transmission percentages of the excitation through the delay shifter before the sample was excited. The power used for the experiment was raised sequentially to 24.8 dBm until the observation of a structural phase transition in the material. The RF power of 24.8 dBm corresponds to the 5.5 V amplitude mentioned in Fig. 2 calculated using a 50 Ω resistance that is present on the chip carrier of the sample holder. It is important to note that due to parasitic reflections, the RF power at the entrance of the sample holder may differ from the power at the sample electrode.

**Imaging conditions.** The time-resolved images were acquired using conventional bright-field imaging at 8kx magnification with the 200 μm condenser aperture and the 60 μm objective aperture on the DECTRIS Quadro direct electron detector over an acquisition time of 5 s. For both MHz and GHz experiments, the Selected Area Electron Diffraction images were recorded using a 10 μm selected area aperture (corresponding to roughly 400 nm in the field of view) with an acquisition time of 1 s. The acquisition times were chosen to balance sample drift during electrical excitation with signal-to-noise ratio. The beam current on the sample is cut from the continuous beam using the 25 μm chopping aperture in the pulser. The beam current densities for the high and low frequency experiments were 12.1 A/m² and 0.36 A/m², respectively.

**Mott Resistor Network simulation**
The Mott Resistor Network (MRN), first introduced in Stoliar et al.[44], describes the VO₂ sample as a two-dimensional grid of cells such that each cell contains four resistors and corresponds to a nanoscale region of the device. The resistors connect each of the cells to their neighbors, and in addition the resistance value of each resistor depends on the local temperature of the cell it belongs. To model the first-order insulator-metal phase transition of VO₂, the stability of each resistor's phase (that is, either insulating or metallic) depends on a Landau-type free energy functional, which is controlled by the local temperature[40]. The cells of the model corresponding to the electrodes are assumed to be ideally metallic, while those corresponding to the vacuum are assumed to be ideally insulating. When applying a voltage across the entire device via the electrodes, current starts to flow through the resistors and locally generates heat according to Joule's first law, P = IV. Each cell of the model is assumed to be in contact with a perfectly insulating substrate that is held to a fixed temperature. For each cell, the local temperature is determined by the sum of the Joule heating contribution, the heat exchanged between neighboring cells, and the heat exchanged with the substrate. Once the local temperature is calculated in each time step, it is used to update the resistance value of the four resistors within the cell. The total resistance of the device collapses when a percolating group of metallic cells forms a highly conducting filament between the two electrodes.

## Data availability

All UTEM videos supporting the results presented in this manuscript are provided in the Supplementary Information. Raw data, prior to video compilation, are available at the following public repository: https://doi.org/10.5281/zenodo.18554592.

## Code availability

The raw data and machine learning code used to compute the phase map in Figure S3 are available at the following public repository: https://doi.org/10.5281/zenodo.14767722.

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

## Acknowledgments

Work was supported as part of the Quantum Materials for Energy Efficient Neuromorphic Computing (Q-MEEN-C), an Energy Frontier Research Center funded by the U.S. DOE-BES, under Award #DE-SC0019273. The development of the microwave-driven pulsed electron microscopy and part of the microscopy analysis efforts (CL, SAR, MGH, LW) at BNL were supported by DOE-BES, Materials Science and Engineering Division, under Contract No. DE-SC0012704. This research used the nanofabrication facility of the CFN, which is a U.S. Department of Energy Office of Science User Facility at BNL and supported under Contract No. DE-SC0012704.

## Author contributions

A.P. initiated the idea. A.P., C.L., S.A.R., S.R., and Y.Z. conceptualized the study. A.P., C.L., S.A.R., M.G.H., L.W., and Y.Z. developed and implemented the ultrafast transmission electron microscopy methods. H.N., E.Q., and T.W.D. performed sample growth, resistance–temperature measurements, and X-ray diffraction characterization. C.L., A.P., and M.G.H. carried out the device fabrication. D.J.A. conducted the Mott resistor network simulations. A.P., S.A.R., S.M., C.L., D.J.A., H.N., E.Q., and T.W.D. contributed to data extraction, analysis, and visualization. M.R., S.R., I.K.S., and Y.Z. secured funding for the project. All authors contributed to the interpretation, discussion and presentation of the results and to the writing, editing, and reviewing of the manuscript.

## Competing interests

The authors declare no competing interests.
