## [Transparent Peer Review file · Nature Communications]

Switching Speed Limits in Electrically Driven VO₂ Structural Mott–Peierls Transition

Corresponding Author: Dr Alexandre Pofelski

Version 0:

Reviewer comments:

Reviewer #1

(Remarks to the Author)

The authors investigated the ultrafast structural dynamics of VO₂ devices under high-frequency electrical excitation using microwave-driven ultrafast transmission electron microscopy (UTEM). They successfully visualized the nucleation, growth, and dissolution of metallic domains in the MHz regime. A key finding is the identification of a frequency limit in the GHz regime, where the material fails to recover its insulating phase due to thermal constraints. This study provides valuable insights into the fundamental "speed limits" governing structural phase transitions in Mott–Peierls materials, and it demonstrates an impressive experimental framework for evaluating and engineering the switching dynamics of Mott-based devices. The revised manuscript more clearly elucidates the distinguishing features and originality of this study, and I believe it now provides sufficient justification for publication. After the authors address the remaining questions noted below, I would recommend the manuscript for publication in Nature Communications.

1) Now I fully understand the challenges associated with developing an experimental setup for performing in-situ electrical measurements concurrently with UTEM imaging. While this limitation is understandable, the conclusion that the device is "locked" in the metallic state at GHz frequencies is supported solely by structural diffraction data. Without electrical response data, the connection between the observed "structural locking" and actual "electrical switching failure" remains elusive. To strengthen this conclusion, the authors should provide electrical characterization of a standard device (fabricated on the same substrate, not the TEM lamella) driven at MHz-GHz frequencies. Demonstrating that the resistance modulation vanishes at 4 GHz in a standard device would render the claims presented in this manuscript considerably more convincing.

2) The authors have shown in Figure 3a that clear nucleation, growth, coalescence and dissolution processes can be resolved on the nanosecond timescale, thereby revealing domain-level switching characteristics. Would it also be feasible to track the evolution of the metallic domain network under GHz electrical pulse excitation, where the relevant dynamics occur much more rapidly? It is critical to distinguish whether the observed behavior truly reflects the fundamental speed limit of VO₂ switching, or whether it instead arises from the formation of a localized, quasi-permanent metallic channel, while some domains may still relax back into the insulating phase.

3) In the revised manuscript, the authors have more clearly delineated the distinctions and advancements relative to prior phase-field simulations under photoexcitation and electrical-thermal models based on DC Mott switching experiments. Given that thermal management—through device-structure engineering, incorporation of metallic NPs, and other strategies for controlling device-level switching dynamics of VO₂—has emerged as a critical topic in recent Mott-material-based neuromorphic and oscillator applications (Nat. Comm. 13. 1 (2022):4609; Nat. Comm. 15.1 (2024):5820), the significance of a high-frequency electro-thermal model combined with TEM imaging analysis is particularly notable. From this perspective, it would be beneficial if the introduction further emphasized the importance of this UTEM technique.

4) The identification of nucleation sites relies heavily on difference imaging and extensive denoising algorithms. There is a risk that aggressive denoising may introduce artifacts that resemble nucleation sites, especially given the weak signal strength. To ensure transparency and reproducibility, it is recommended that the authors provide a supplementary figure illustrating the step-by-step image processing workflow. This figure should visually present the evolution of a representative

frame through the pipeline: (1) the raw difference image, (2) the image after denoising/filtering, and (3) the final binary mask after thresholding. In addition, applying this same pipeline to a “null” dataset (e.g., RF-off or pre-t0 frames) would help demonstrate that the algorithm does not generate false-positive signals in the absence of excitation.

5) In Figure 3a, during the structural switching, significant contrast fluctuations (ripples) are observed in the Al₂O₃ substrate. While this is likely due to strain-induced bending contours arising from the volume expansion of VO₂, it is not explicitly explained in the text or caption. If the authors also discuss the stress transfer induced by the structural phase transition of VO₂ in Figure 3a, along with the subsequent mechanical responses of the substrate, it would provide readers with additional insight and valuable clues regarding the underlying dynamics.

Reviewer #2

(Remarks to the Author)

The revised manuscript by Pofelski and co-workers properly addresses my questions and comments raised regarding the original submission for Nat. Nanotech.. As I have already written in the previous review report, I believe the manuscript reports solid and interesting results. In terms of level of novelty and insights, complexity of the investigation and thorough analysis, Nature Communication seems to me a fitting Journal and I recommend publication in its current form.

Version 1:

Reviewer comments:

Reviewer #1

(Remarks to the Author)

I have no additional comment. This nice work can be published in Nature Communications.

Response to reviewer #1:

The authors investigated the ultrafast structural dynamics of VO₂ devices under high-frequency electrical excitation using microwave-driven ultrafast transmission electron microscopy (UTEM). They successfully visualized the nucleation, growth, and dissolution of metallic domains in the MHz regime. A key finding is the identification of a frequency limit in the GHz regime, where the material fails to recover its insulating phase due to thermal constraints. This study provides valuable insights into the fundamental "speed limits" governing structural phase transitions in Mott-Peierls materials, and it demonstrates an impressive experimental framework for evaluating and engineering the switching dynamics of Mott-based devices. The revised manuscript more clearly elucidates the distinguishing features and originality of this study, and I believe it now provides sufficient justification for publication. After the authors address the remaining questions noted below, I would recommend the manuscript for publication in Nature Communications.

1) Now I fully understand the challenges associated with developing an experimental setup for performing in-situ electrical measurements concurrently with UTEM imaging. While this limitation is understandable, the conclusion that the device is "locked" in the metallic state at GHz frequencies is supported solely by structural diffraction data. Without electrical response data, the connection between the observed "structural locking" and actual "electrical switching failure" remains elusive. To strengthen this conclusion, the authors should provide electrical characterization of a standard device (fabricated on the same substrate, not the TEM lamella) driven at MHz-GHz frequencies. Demonstrating that the resistance modulation vanishes at 4 GHz in a standard device would render the claims presented in this manuscript considerably more convincing.

Author response: We thank the reviewer for this valuable suggestion. We recognize the value of performing MHz–GHz electrical characterization on a standard device fabricated on the same wafer. Unfortunately, we do not have the capability to perform high frequency measurements on our probe station due to impedance-matching constraints (essential for high frequency measurements) and the intrinsic RC limit from the experimental setup presently. While this prevents direct GHz electrical measurements, we have performed complementary time-resolved pulsed electrical experiments designed to probe the VO₂ device relaxation that are directly relevant to high-frequency operation.

Specifically, we have performed time-resolved pulsed electrical measurements, presented in Fig R2-1, at 20 kHz on a VO₂ device fabricated on the same sapphire substrate. A pulse train is sent on the electrical circuit with variable inter-pulse intervals as shown on the left panel of Fig R2-1. The inter-pulse interval allows the device to relax electrically to its high resistivity state before the next pulse is initiated. By progressively decreasing the inter-pulse interval, we probe how electrical switching is preserved by monitoring the threshold voltage extracted from time-resolved voltage measurements on VO₂. The threshold voltage is indeed reduced if the VO₂ device is partially relaxed before the next pulse.

The right panel in Fig R2-1 shows that the threshold voltage depends strongly on the inter-pulse interval. When two pulses are separated by several microseconds, the device relaxes thermally and the threshold voltage recovers to its original value (~5.2 V). However, as the inter-pulse interval decreases toward the sub-microsecond regime, the threshold voltage drops markedly (to ~4.0 V), indicating that the device no longer returns to the fully insulating state between pulses.

Fig R2-1: Time-resolved pulsed electrical measurement of a VO₂ device at 20 kHz. Left: Electrical measurement configuration showing the applied voltage pulse train (black dashed line) and the measured voltage response V1 across the device (red). Right: Extracted threshold voltage of the VO₂ device as a function of inter-pulse interval. For long inter-pulse intervals, the device fully relaxes to the high-resistivity monoclinic state, and the threshold voltage recovers to its nominal value. As the inter-pulse interval decreases, incomplete relaxation leads to a progressive reduction of the threshold voltage, indicating partial retention of the metallic state. Extrapolation toward shorter time intervals suggests that, at sufficiently high frequencies (above GHz), the threshold voltage effectively vanishes, leaving the device locked in a metallic state.

Extrapolating the experimentally observed relaxation trend in Fig R2-1 towards higher frequency, a 4 GHz excitation corresponds to a 0.25 ns interval, which is more than four orders of magnitude shorter than the thermal cooling time extracted from our data of several microseconds. At such short intervals, the device cannot recover its insulating state, and therefore electrical resistance modulation necessarily collapses locking the device in its metallic state. In other words, the absence of electrical switching at GHz frequencies is a consequence of incomplete thermal relaxation that matches the UTEM structural observation.

This behavior is consistent with thermally assisted locking phenomena that emerge at higher repetition rates. We interpret the structural locking observed under GHz excitation in UTEM as the structural manifestation of the electrically metallic locked regime, where thermal dissipation becomes insufficient to restore the monoclinic phase between excitation cycles.

To address the reviewer’s comment, we made the following changes in the manuscript and added Fig. R2-1 to the supplementary information as Fig. S10:

“Additional electrical measurements performed at moderate frequency on a VO₂/sapphire device in planar configuration are presented in Supplementary Fig. S10 and confirm the impact of the thermal budget on the recovery of the high-resistivity state (monoclinic phase). These measurements further demonstrate that high-frequency cycling is electrically limited by the metallic dissolution process, emphasizing the critical role of pulse shape and temporal profile when increasing the operating frequency as suggested by the MRN model.”

2) The authors have shown in Figure 3a that clear nucleation, growth, coalescence and dissolution processes can be resolved on the nanosecond timescale, thereby revealing domain-level switching characteristics. Would it also be feasible to track the evolution of the metallic domain network under GHz electrical pulse excitation, where the relevant dynamics occur much more rapidly? It is critical to distinguish whether the observed behavior truly reflects the fundamental speed limit of VO₂ switching, or whether it instead arises from the formation of a localized, quasi-permanent metallic channel, while some domains may still relax back into the insulating phase.

Author response: We thank the reviewer for this important question, which addresses the distinction between reversible domain-level switching and irreversible metallic channel formation at high excitation frequencies. Here, the term reversible refers to the ability of the VO₂ film to recover its monoclinic state within the period of the electrical excitation. As shown in Figure 2 of the main manuscript, the VO₂ device switches between metallic and insulating states at 1 MHz but becomes locked in the metallic state at 4 GHz, with no reversal to the monoclinic phase over the RF period. A similar phenomenon is observed in Figure 4, where increasing the pulse duration beyond a critical threshold locks the VO₂ device into the metallic state. Imaging reversible insulating/metallic switching and imaging the locking to the metallic state require two distinct experimental approaches.

Our ultrafast TEM measurements are performed in a synchronized pump–probe configuration, which inherently restricts observation to repetitive and reversible dynamics. In such stroboscopic experiments, the electron probe is synchronized to a fixed delay t within each excitation cycle, effectively “freezing” the sample state at that time point over many identical cycles. For example, a one-second acquisition at 1 MHz corresponds to the cumulative integration of one million identical switching cycle “frozen” at the same precise time point t . For the cumulative recording to be coherent with the sample dynamics, the sample must return to the same microscopic state at each cycle. Otherwise, the resulting image would blur due to temporal averaging over different sample states. In the case of irreversible processes, only the final state is recorded, as the dynamics occur solely during the first cycle and cannot be captured stroboscopically. Consequently, the successful acquisition of time-resolved UTEM image series, like in Figure 3, itself constitutes strong evidence of cycle-to-cycle reversibility of the sample dynamics.

Under GHz electrical excitation, the VO₂ device does not exhibit reversible structural cycling. As a result, the associated structural dynamics are non-repetitive and irreversible on the timescale of the excitation, making them incompatible with synchronized pump–probe UTEM imaging. Capturing the transient evolution of irreversible dynamics requires a different experimental approach, such as direct imaging under in-situ high frequency electrical biasing using a high framerate detector or Dynamic Transmission Electron Microscopy (DTEM) using a beam blanker synchronized with the pump. At present, however, no electron microscopy instrumentation exists that can directly image irreversible structural processes under GHz electrical excitation in a non-stroboscopic manner. Current DTEM implementations rely on laser-based excitation, and available direct electron detectors do not reach frame rates in the GHz regime. Additional challenges related to electron beam brightness and dose efficiency further complicate such measurements. UTEM imaging of irreversible GHz switching dynamics therefore remains an important experimental instrumental development field that is widely open for future exploration.

3) *In the revised manuscript, the authors have more clearly delineated the distinctions and advancements relative to prior phase-field simulations under photoexcitation and electrical-thermal models based on DC Mott switching experiments. Given that thermal management —through device-structure engineering, incorporation of metallic NPs, and other strategies for controlling device-level switching dynamics of VO₂ — has emerged as a critical topic in recent Mott-material-based neuromorphic and oscillator applications (Nat. Comm. 13. 1 (2022):4609; Nat. Comm. 15.1 (2024):5820), the significance of a high-frequency electro-thermal model combined with TEM imaging analysis is particularly notable. From this perspective, it would be beneficial if the introduction further emphasized the importance of this UTEM technique.*

Author response: We thank the reviewer for highlighting these recent studies and for emphasizing the growing importance of thermal management in Mott insulator devices operating at high frequencies. We agree that device level thermal engineering has emerged as a critical parameter for controlling switching dynamics and electronic functionalities in VO₂ based neuromorphic and oscillator architectures.

In this context, electrically driven UTEM is particularly well suited to probe such effects, as it provides direct access to structural dynamics under high-frequency electrical excitation and is applicable to complex device geometries and engineered architectures. While the scope of the present work is not limited to specific thermal management strategies, our approach establishes a general experimental framework to visualize and quantify electrically driven structural dynamics at the relevant spatiotemporal scales.

To address the reviewer's comment, we have revised the Introduction as follows:

“Numerous studies demonstrate the potential of VO₂ as high-frequency switches, neuronal oscillators, and sensors justifying the strong and broad interest of the research community^{29,30}. More recently, device-level thermal coupling strategies have been shown to modulate electronic functionalities, further enhancing the potential of VO₂-based devices^{31–33}. Nonetheless, there is a knowledge gap regarding the structural information and dynamics of quantum materials such as vanadium dioxide operating under high-frequency electrical excitation.”

4) *The identification of nucleation sites relies heavily on difference imaging and extensive denoising algorithms. There is a risk that aggressive denoising may introduce artifacts that resemble nucleation sites, especially given the weak signal strength. To ensure transparency and reproducibility, it is recommended that the authors provide a supplementary figure illustrating the step-by-step image processing workflow. This figure should visually present the evolution of a representative frame through the pipeline: (1) the raw difference image, (2) the image after denoising/filtering, and (3) the final binary mask after thresholding. In addition, applying this same pipeline to a “null” dataset (e.g., RF-off or pre-t₀ frames) would help demonstrate that the algorithm does not generate false-positive signals in the absence of excitation.*

We thank the reviewer for the comment allowing us to provide more details. To clarify, no denoised data is presented in the main manuscript. We intentionally adopted a conservative processing strategy to remain as close as possible to the raw experimental signal. The processing pipeline consists of (i) rigid image registration to correct for sample drift, (ii) difference imaging relative to a

reference frame, (iii) contrast adjustment, and (iv) a gentle Gaussian smoothing. This approach lets the reader being guided through responsible data visualization.

As the data remained noisy, additional denoising, thresholding, and filtering steps were applied in the supplementary information only to aid the interpretation of the data presented in the main manuscript. While a limited number of false positives are noticeable after thresholding the denoised data in the supplementary figures S6 and S7, the structural phase distribution remains consistent with the image-difference contrast at different time frame of the VO₂ IMT, facilitating thus the reading of the figures in the main manuscript.

To address the reviewer’s comment, we added the following Fig. R2-2 as Fig. S6 in the supplementary information highlighting the entire processing workflow focusing on the nucleation site of the VO₂ device at representative time points of the structural cycle. We believe this added figure provides sufficient information to assess the robustness of the processing workflow and ensure reproducibility.

Figure R2-2: Step-by-step image processing workflow focusing on the right-hand side nucleation site of the VO₂ device on representative time points of the 1 us cycle. For all data reported in the main manuscript, the raw images were first rigidly registered to correct for sample drift. Difference images were then computed relative to a reference frame acquired at t = -70 ns, adding a gentle Gaussian smoothing and contrast adjustment to enhance visibility of weak structural contrast changes. The

time $t = 0$ was defined as the moment when the contrast variation at the nucleation site exceeded the noise level in the difference image. For the supplementary information only, the difference image series was further processed using denoising, thresholding, and filtering steps to generate binary masks labeling insulating and metallic regions, which serve as a guide for data interpretation.

5) In Figure 3a, during the structural switching, significant contrast fluctuations (ripples) are observed in the Al₂O₃ substrate. While this is likely due to strain-induced bending contours arising from the volume expansion of VO₂, it is not explicitly explained in the text or caption. If the authors also discuss the stress transfer induced by the structural phase transition of VO₂ in Figure 3a, along with the subsequent mechanical responses of the substrate, it would provide readers with additional insight and valuable clues regarding the underlying dynamics.

Author response: We thank the reviewer for the comment. The contrast fluctuations in the substrate indeed correspond to motions of bending contours demonstrating a structural distortion (subtle deformation and/or rotation) of the sapphire following the structural dynamics of the VO₂ material. The change of volume of VO₂ during the structural transition is imposing mechanical constraints that are propagating down to the substrate.

In the present study, our primary focus is on the structural phase evolution within the VO₂ film itself. Consequently, the information extracted from the substrate response remains qualitative. The experiment was designed to capture the ultrafast structural dynamics of VO₂, and the sensitivity to substrate deformation became evident upon post-processing of the data. While this limits a quantitative strain analysis in the sapphire, the observed bending-contour dynamics nonetheless provides useful qualitative insight into film-substrate coupling during switching and into the physical mechanism supporting the propagation of the structural wavefront.

To address the reviewer's comments, we have expanded the discussion associated with Figure 3 to describe the origin and implications of the substrate ripples, as detailed below.

“Additional insight can be obtained from the substrate response itself, as ripple-like contrast fluctuations are visible in the sapphire region in Fig. 3a, concomitant with the structural dynamics of the VO₂ film. These contrast variations arise from the motion of bending contours, indicating elastic distortions of the substrate induced by the VO₂ structural phase transition. As VO₂ undergoes a slight volume change upon switching to the metallic phase, a mechanical constraint is imposed on the underlying sapphire, which is expected to respond elastically. No measurable delay is observed between the nucleation of the metallic phase at the top of the VO₂ film and the initiation of bending contour motion in the substrate. This suggests that, within the temporal resolution of the experiment (2.2 ns), mechanical stress propagates from the nucleation site into the substrate without inducing a structural transformation of the VO₂ region in between. While this observation confirms the stress transfer across the film-substrate interface, it also indicates that mechanical contributions are not the dominant factor governing the propagation of the IMT structural front in VO₂.

Response to reviewer #2:

The revised manuscript by Pofelski and co-workers properly addresses my questions and comments raised regarding the original submission for Nat. Nanotech.. As I have already written in the previous review report, I believe the manuscript reports solid and interesting results. In terms of level of novelty and insights, complexity of the investigation and thorough analysis, Nature Communication seems to me a fitting Journal and I recommend publication in its current form.

Author response: We thank the reviewer for the positive assessment and constructive feedback through the review process, and we are grateful for the recommendation for publication in *Nature Communications*.

Response to reviewer #1:

Reviewer #1 (Remarks to the Author):

I have no additional comment. This nice work can be published in Nature Communications.

Author response: We thank the reviewer for the constructive feedback through the review process, and we are grateful for the recommendation for publication in *Nature Communications*.